# Factors Associated with Cumulative First-Week Mortality in Broiler Chicks

**DOI:** 10.3390/ani10020310

**Published:** 2020-02-17

**Authors:** Marta Yerpes, Pol Llonch, Xavier Manteca

**Affiliations:** Animal Nutrition and Welfare Service, Department of Animal and Food Science, School of Veterinary Science, Universidad Autónoma de Barcelona, 08193 Barcelona, Spain; Pol.Llonch@uab.cat (P.L.); Xavier.Manteca@uab.cat (X.M.)

**Keywords:** broiler, first week mortality, welfare, performance, management, factors

## Abstract

**Simple Summary:**

The first week of life of broiler chicks is a sensitive period where many of the chicks’ systems and organs are considered immature. During this period there are a lot of factors that can negatively influence chick morpho-physiology affecting welfare. A decrease in early life chick welfare could be reflected in first-week mortality. In this study we used data from one hatchery company to identify risk factors that could influence first-week mortality. Risk factors were classified either as internal (individual-dependent) and external (management or environmental) factors. We found that breeder age, chick gender and breed were the internal factors significantly related to chick mortality. Among the 21 external factors considered only type of broiler house, presence or absence of drip cup, egg storage, study year and season were related to chick mortality. In conclusion, the identified housing factors and management routines should be considered to reduce first-week mortality rate, to rearing the chickens and to the construction of new broiler houses.

**Abstract:**

First-week mortality is an important performance index as well as an important welfare indicator. The aim of the present study was to identify internal (individual-dependent) and external (management or environmental) factors that could influence the cumulative first-week mortality of broilers. To carry out this study, field data obtained from a hatchery company were used, in which 2267 flocks of broiler chicks (from 2015 to 2018), were analyzed. A generalized linear mixed model was used to analyze the data. Farm ID and house by farm were incorporated as random effects. The Odds Ratio was estimated for each factor, determining the effect of each explanatory variable. First-week mortality was significantly related to breeder age (*p* < 0.0001), chick gender (*p* < 0.0001) and breed (*p* < 0.0001) as internal factors, and type of broiler house (*p* = 0.0129), presence or absence of drip cup (*p* < 0.0001), egg storage (*p* < 0.0001), study year (*p* < 0.0001) and season (*p* < 0.0001) as external factors. Therefore, these factors should be considered in the decision making of poultry breeding companies, in order to reduce possible welfare problems and increase productive performance.

## 1. Introduction

In current broiler production systems, the first week of the chicks’ life represents between 16 to 20% of their productive life. These first days of life are a transition stage in which many things happen in a very short period of time. First, chicks change from a very conditioned and controlled life in the hatchery to a more independent life in the farm [1]. At this transition stage, many of the chicks’ internal systems are immature. Hence, it is during this sensitive stage that the greatest changes will occur requiring an adaptation time [2]. Therefore, during this first week of life chicks are submitted to high degree of stress as they try to adapt to all these challenges, and maximize their chances of survival. All these factors that can negatively influence chick morpho-physiology during this stage will have an impact on welfare and, if animals fail to adapt, could lead to increased mortality in the first week.

Mortality during the first week of life in broilers is an important production criterion, widely used in poultry production. It is also one of the animal welfare indicators covered by the 2007/43/EC European Directive [3]. In addition, first-week mortality is a common parameter used to assess chick quality. Chick quality can be defined as chicks that are optimally developed during incubation and show a high-performance potential and survivability [4]. Expanding on this concept, Ulmer-Franco et al. [5] described that chicks hatched with good quality are those that are active, have a body weigh between 40–44 g, their navel is healed upon removal from the hatcher, and their down is dry. Importantly, the quality of day-old chicks is directly related to the survival rate at the first week of life [1]. This measure is useful to provide information about the breeder farm, the hatchery and the individuals. Chick viability and chick quality depend on internal (individual-specific) and external (management or environmental) factors. Breed and breeder age can affect chick quality and in consequence the performance and welfare of the chicks [2,6,7]. There are also external factors that can influence first-week mortality such as season [8], flock size, density of chicks housed in the farm, type of ventilation and type of drinkers [1,9,10]. For instance, according to Chou et al. [10], an increase of the size of the flock decreases the cumulative first-week mortality.

Previous studies that assessed risk factors of first week mortality, included only a limited number of them. However, the present longitudinal study enlarges this list including an extensive number of factors (up to 24) related to chick, environment and management. The aim of this study was to identify internal and external factors that could influence the cumulative first-week mortality of broilers.

## 2. Materials and Methods

### 2.1. Data Collection

For the study, information related to first-week mortality was requested from a Spanish broiler company, located in Zaragoza (Spain). As well as information on the characteristics of farms, the company provided the following databases:A descriptive database of the different farms that belongs to the company. In which characteristics are described for 253 broiler houses of 104 farms.A database including the first-week mortality of chicks during the last four years: 2015, 2016, 2017 and 2018. This database contains 6979 mortality records, including the flock, gender and broiler house (or half broiler house). A total of 2267 flocks, 70,211,678 day-old chicks and 68,949,774 seven-day old chicks were studied.

In the study conducted by Yassin et al. [1] the data included information from years 2004, 2005 and 2006 from two hatcheries and 482 broiler farms; they also included in their analysis data from 16,365 flocks. Heier et al. [9] used data from 1996 to 1999 from 1664 flocks in 132 farms, whereas Chou et al. [10] used data from year 2000 in a study of 68 hatcheries and 4796 flocks in 848 farms. As in our study, all data of the mentioned studies were collected and recorded by the companies involved.

### 2.2. Factors under Study

To carry out the study, 24 variables were considered as risk factors in chick mortality. These variables were classified as internal and external factors, which are described in Table 1.

#### 2.2.1. Internal Factors

Three variables were considered factors specific to the individual: breeder age—intervals expressed in weeks and categorized according to the Cobb guide [11] (25–33 weeks, 34–50 weeks and 51–68 weeks and unknown), chick gender (male, female and mixed) and breed (Cobb and Ross). These internal factors were chosen due to the relationship detected in other studies with the viability and chick quality [1,2,6].

#### 2.2.2. External Factors

There were 21 variables classified as environmental or management factors. The factors under study are listed in Table 1, except for the continuous variable constructed area (m^2^). It should be noted that the variables ‘farm entry density’ (chicks/m^2^) and ‘flock size’ (considering the number of chicks received) were categorized according to 25th, 50th and 75th percentiles. These external factors were considered because there is evidence showing its importance on the viability of the chicks. Factors such as season [8], transport [10], stocking density, flock size, feeding and drinking system, ventilation and insulation at the broiler farm [9] were related to first week mortality. In addition, Yassin et al. [1] detected that egg storage days was negative related with first week mortality and there was a significant difference in first week mortality in the different study years.

### 2.3. Statistical Analysis

Statistical analysis was carried out using the statistical software SAS v9.4, (SAS Institute Inc., Cary, NC, USA) for Windows. The significance level was set at 0.05 and the study unit was the flock. According to the graphical evaluation of the residues (histogram and quantile-quantile plot), the analyzed data was not normally distributed.

The cumulative first-week mortality (FWM) was calculated as a percentage. The total number of dead chicks plus discarded chicks during the first week of life was used as a numerator, and the number of chicks received in the farm as a denominator. Criteria for excluding chicks from the study were the same for all farms; chicks that showed clinical signs of severe disease, abnormalities or chicks with inequality in growth (little chicks).

Secondly, a univariate descriptive analysis between mortality at first day of life and FWM was carried out. A bivariate linear regression analysis was performed to examine the relationship between the explanatory variables collected and the response variable FWM. The correlations study was conducted using the Spearman correlation coefficient. In order to detect risk factors related to FWM, a generalized linear mixed model was used using a binomial distribution (number of chicks dead + discarded chicks at first week/number of total chicks received at the farm). ‘Farm ID’ and ‘broiler house’ by farm were incorporated in the model as random effects. All significant explanatory variables detected in the bivariate analysis were included in a preliminary multivariable model. The variables were removed manually one by one from the model using a significance level of 0.05. For each factor, the objective was to estimate the probability of mortality (p) using Odds Ratio (OR) (p/(1 − p)), determining the effect of each explanatory variable.
$$\ln\left(\frac{p}{1-p}\right)=\beta_{0}+\underset{i}{\sum }\underset{j}{\sum }\beta_{i}*\left({\mathrm{Variable}}_{i}={\mathrm{Category}}_{j}\right)$$
each parameter was estimated (β) considering:Variable_i_ = each of the study factor (explanatory variables).Category_j_ = each category of the explanatory variable.

For each significant variable, the estimation of the proportion and confidence intervals was calculated. The confidence interval was set at 95%, which was intended to estimate a range of plausible values for an unknown parameter (in this case mortality risk ≈ OR) and these intervals have an associated confidence level (0.05) where the true parameter is in the proposed range. We analyzed the categories from each factor, comparing categories and obtaining the mortality risk, which is equivalent to OR.

## 3. Results

The FWM of the studied flocks ranged from 0.13% (0.05th percentile) to 2.25% (0.95th percentile) with an average of 1.82% (±0.99%). The generalized linear mixed model identified the following factors as significant (*p* < 0.05): breeder age, chick gender, breed, type of broiler house, presence of drip cup, egg storage, study year and season (Table 2). External factors related to the environment (ventilation or type of insulation), feeding and bedding did not have a significant influence on FWM.

To determine the effect of each explanatory variable, Table 3 shows the results obtained in the OR magnitude between each pair of categories. A one by one analysis for every factor was performed, in order to obtain the first-week mortality risk depending on the category.

According to our results, chicks housed in farms with one floor had a higher FWM (1.83%) and had 9% more mortality risk than those chicks housed in multiple floors (1.75%). The FWM was higher in chicks from old breeders (51–68 weeks) (2.10%) compared with chicks from young (25–33 weeks) (1.79%) and prime breeders (34–50 weeks) (1.76%). In addition, chicks from old breeders had a 26% and 22% higher mortality risk during the first week of life than chicks from young and prime breeders, respectively. Irrespective of differences in mortality risk it was considered to be low (4%). Regarding chick gender, the FWM was higher for mixed (1.91%) and male chick flocks (1.89%) compared with female chick flocks (1.74%). In fact, male and mixed chick flocks had a mortality risk 11% and 17% higher than females, respectively. In the case of breed, Ross chicks had a higher FWM (1.85%) than Cobb chicks (1.72%). With regard to mortality risk, chicks from Ross breed had a mortality risk 8% higher than chicks from Cobb breed. In the case of chicks housed on farms without presence of drip cup, the FWM was higher (1.98%), as it was the mortality risk, which was 17% higher than chicks housed on farms with presence of drip cups (1.66%). The FWM was higher in chicks from eggs stored less than 7 days (1.94%) compared with chicks from eggs stored from 7 to 10 days (1.76%) and eggs stored more than 10 days (1.76%). The mortality risk was higher in chicks from eggs stored less than 7 days. Chicks from eggs stored less than 7 days had 7% higher mortality risk compared to 7 to 10 days storage. Chicks from eggs stored for more than 10 days had a 10% lower mortality risk than 7 days storage. Regarding seasonality, autumn and winter had the highest FWM (1.86% and 1.93%, respectively) compared with spring and summer (1.69% and 1.81%, respectively). Equally, the mortality risk during first-week of life was higher in autumn and winter compared to spring and summer. In winter, the mortality risk was 10% higher than in spring, 5% higher than summer and 1% higher than autumn. In autumn, the mortality risk was 8% higher than spring and 3% higher than summer. In addition, chicks reared in summer had a 5% higher mortality risk than chicks reared in spring. Regarding the study year, 2015 had the highest FWM (1.96%) compared with 2016 (1.75%), 2017 (1.80%) and 2018 (1.79%). Similar to these results, the highest mortality risk was in 2015. In 2015 the mortality risk was 9% higher than in 2016, 6% higher than in 2017 and 10% higher than in 2018. In addition, the mortality risk increased by 2% and 3% in 2017 compared to 2016 and 2018, respectively. Similarly, the mortality risk in 2016 was higher (1%) than in 2018.

## 4. Discussion

At the farm, the weekly mortality rate changes over time; according to Heier et al. [9] the average of FWM was 1.54% and 0.48% per week during the rest of the growth period. According to Chou et al. [10], the average FWM was 1.32% and according to the results of Awobajo et al. [14], the average FWM was 1.7%. Therefore, the results of the mean FWM obtained in Awobajo’s study seem to be in line with the findings observed in the present study.

All internal factors and five out of the 21 external factors considered in the present study were significantly related to FWM, which are discussed in the following section. Among the external factors that did not affect FWM, there were ventilation and air quality, litter quality and feeding.

Ventilation was also shown to be independent to FWM in studies conducted by Heier el al. [9] and Chou et al. [10]. This may be due to the fact that the fresh air requirement during the first days of life is very low and there is no significant accumulation of ammonia. Therefore, the necessity for using active air supply in this period is questionable.

The fact that mortality during the first days and litter quality were not related was also found by [12,15]. However, we envisage that if there is a problem with the water supply system or if a number of factors combine (for instance old facilities, unshredded straw litter, or/and inadequate insulation) it could result in litter wetting, and higher mortalities could happen due to a lower capacity of chick to maintain body temperature.

In relation to feeding, no significant effect on FWM was detected in our study in contrast to other studies, where it was found that the presence of feed on paper and the feed-providing companies of the breeder affected FWM [1,9]. The reason for differences in other studies cannot be inferred from the data provided, but from our results, feed does not affect FWM. In contrast to our results, Heier et al. [9] detected that the insulation of the farm had an influence on FWM. In their study, they analyzed the presence or absence of floor insulation whereas we analyzed the types of insulation, which might indicate that it is more important to have insulation than the type of insulation. Finally, it is important to emphasized that one of the most important factors during the first days of brooding are ambient and litter temperature. Authors demonstrated that lower brooding temperatures influence mortality and production indexes. However, ambient and litter temperature have already received significant attention in previous studies [16,17,18] and was deemed unnecessary to persevere on them.

### 4.1. Internal Factors

#### 4.1.1. Breeder Age

Yassin et al. [1] and Peebles et al. [6] claimed that there is a close relationship between FWM and breeder age. According to our results, the older the breeders, the higher both the FWM rate and mortality risk during the first week of life. In contrast to our results, other studies suggest that the highest rate of FWM occurs in chicks from young breeders (25–33 weeks), followed by chicks from old breeders (51–68 weeks) [19,20]. These differences might be because these studies only used two of the three breeder ages used in the present study. Wilson [19] used chicks from young and old breeders, and Suarez et al. [20] used chicks from young and prime breeders.

The explanation for the results obtained in the present study could be due to the correlation between breeder age with egg size and incubation time [19,20,21]. Due to the interaction between breeder age and egg size, eggs from young breeders need more hours of incubation compared with eggs from prime and old breeders [20]. As a result, eggs from old breeders hatch earlier [20], and therefore, the risk of chick dehydration increases with breeder age. Dehydration of chicks at an early age may result in increased FWM [22,23]. Another possible hypothesis is that chicks from old breeders have low chick quality and viability and a worse healing navel, increasing the probability of suffering diverse pathologies and infections of the yolk sac, which might increase FWM [1,22].

Therefore, in everyday hatchery practice, breeder age should be an essential factor for the management of the hatching window and the removal decision of the chicks from the hatcher. In order to improve the welfare of the chicks, breeder age should be considered a major factor in any decision related to management or production, since the negative effect of this factor persists during the performance of the chicks until the end of the growth period [6,24].

#### 4.1.2. Chick Gender

Female chicks obtained a lower FWM and a higher probability that this rate would decrease, compared to male and mixed chicks. The differences observed for males and females would be in line with the results obtained in the study by Leitner et al. [25]. From week two to week eight of life, they observed large differences in the mortality rate between males and females. These differences were found in the activity of the T and B lymphocytes, despite the participation of other regulatory cells could not be ruled out. They concluded that females developed effective immune responses before than males. This difference in the rate of development of immune response is what could make males more susceptible to pathogens, and therefore, have a higher FWM.

However, Wu et al. [26] obtained opposite results for the mortality rate during embryonic development. Wu et al. [26] suggested that the possible causes for these differences in embryo mortality rate could be related to significant differences in embryonic development between males and females, leading to this bias in premature death. Another cause they suggested was that lethal sex-linked genes could account for the differences in early mortality observed between the sexes. The last cause they consider is related to the environment temperature to which embryos are exposed during incubation. Male embryonic mortality is higher at high temperatures, while female embryonic mortality is higher at low temperatures, with similar mortality in both sexes when incubation temperatures are intermediate.

Therefore, in order to reduce FWM, two separate strategies should be considered. One for the embryonic period and another for the postnatal period. During the embryonic period, an intermediate and homogeneous incubation temperature should be ensured to avoid possible embryonic thermal stress. During the postnatal period, adequate vaccination of all chicks should be ensured with special emphasis on male chicks. In this case, the potential cost of implementing the suggested measures would vary depending on the incubators, hatchers and vaccination system used in the hatchery. Nevertheless, due to the high risk of mortality according to the chick gender, the implementation of these measures would be cost-effective in the short term. Finally, in order to confirm these recommendations, further experimental trials would be necessary.

#### 4.1.3. Breed

Chicks from Ross breed obtained a higher FWM, and a higher probability of increasing this rate compared to Cobb breed chicks. These findings would agree with the results obtained in other studies in relation to differences between breeds [1,14]. However, the breeds used to carry out the comparison were different in all the studies. The International Agriculture Report (1994) explains that some breeds are prone to higher FWM due to genetic factors. In fact, Awobajo et al. [14] observed that this difference in mortality rate depending on breed was maintained throughout the entire growing period. Some breed-related factors influencing chick viability and quality and consequently FWM are the difference in egg weight [27] and the daily metabolic activity of the embryo during incubation [28]. As these breed-related factors influence chick quality, hatchery managers should take them into account when deciding which eggs to place together in each hatchery. Therefore, it would be important to consider the attributes of the breed before selecting the one that best suits the needs of each company, location and type of production.

### 4.2. External Factors

#### 4.2.1. Type of Broiler House

According to the results, chicks housed in broiler farms with multiple floors had a lower FWM than chicks housed in broiler farms with one floor. Although farms with multiple floors tend to be older, and therefore, more complicated to manage [29], these types of facilities may be a good option because when broiler houses are preheated to receive chicks, temperature tends to rise, producing a comfortable litter temperature in the upper floors. In line with this hypothesis, Deaton et al. [16] explain that chicks raised at low temperatures have a higher mortality, mainly due to ascites.

In light of these results, we hypothesize that FWM would be more related to the temperature of the house than to its space distribution. Therefore, at an early age, it should be ensured that the house reach an adequate environmental and litter temperature, and that it is homogeneous throughout the housing facilities.

#### 4.2.2. Drip Cup

We found that chicks raised without drip cups had a higher FWM than chicks raised without, and a higher mortality risk during the first week of life. There are currently several water supply systems at farms. However, the most widely used in the Spanish poultry industry is the nipple system, with or without drip cup (drip cup is a part of nipple drinkers to avoid wet litter). The main reasons for its widespread use are the lower labor time in cleaning bell drinkers [30], the improvement in the microbiological quality of the water [31], and the increase in the control of litter moisture [32].

However, this system alters the drinking behavior of poultry, since it is very different from that which would naturally take place. In particular, ‘spooning’, in which the bird lowers its head, takes water in its beak and then raises its head again, is completely absent [33]. In addition, birds have to learn a necessary action for drinking, which is to apply pressure with the beak on the nipple [34]. Despite the important management reasons for choosing this type of system, it does not seem to be the best option for chicks. In addition, this inconsistency with their natural behavior (spooning), a study by Heier et al. [9] comparing different water supply systems, showed that chicks raised with nipple drinkers had a higher FWM. Carpenter et al. [35] compared performance indices in chicks raised with nipple drinkers or with nipple drinkers supported by supplementary drinkers (plate type). They detected that mortality decreased in the latter case.

All the previous literature indicates two possible reasons why chicks reared with nipple drinkers without drip cups had a higher probability of mortality. The first is that having this part of the drinker to collect water in a small plate gives chicks an easy and quick access to the water, thus, they do not need to learn how to drink in a new system. The second reason is that nipple drinkers without drip cup (lacking this water collection system) will be more susceptible to diseases associated to litter humidity. Therefore, the results of this study suggest that the addition of the drip cup may improve the welfare of chicks by reducing the likelihood of dehydration and maintaining litter quality.

#### 4.2.3. Egg Storage Days

Chicks from eggs stored for less than seven days had a higher FWM, and also a higher mortality risk during the first week of life. These results contrast with the findings of Yassin et al. [1], where the FWM increased slightly for each additional day of storage.

It should be noted that egg storage is a common practice in hatcheries in order to synchronize the different activities of the hatchery, such as egg reception, egg sanitization, preheating and schedule incubator and hatcher loading.

Egg storage for up to seven days has little or no effect on hatchability [36]. However, it is well documented that prolonged storage periods decrease hatchability and increases incubation times [37,38], and decreases chick quality [20,39]. It has been suggested that this prolonged storage of the egg may induce embryonic stress, resulting in irreparable damage to the embryo, increasing embryonic mortality and decreasing performance [38,40]. Because of this, various interventions have been developed to decrease the negative effects of prolonged egg storage. One of the most recently applied techniques is the use of short periods of incubation during egg storage (SPIDES). Several studies confirm that this technique provides a good restoration of hatchability and chick quality [41,42,43]. Moreover, Dymond et al. [36] reported that the performance during the first week of life in terms of chick body weight was greater using SPIDES.

The company in this study uses the SPIDES technique routinely, since it is usually applied to eggs stored for more than five days. According to our results, it is reasonable to expect that the FWM would be similar to that typically found with eggs stored for less than seven days. In fact, breeding companies are suggesting to treat eggs when arriving in the hatchery in order to homogenize the temperature and embryonic phase. It could be the reason why the eggs stored for less than seven days obtained these results. Therefore, the results of this and other studies suggest that prolonged egg storage would not have a negative effect on performance, as long as the SPIDES technique is applied.

#### 4.2.4. Study Year

Mortality during the first week of life was significantly different between the different years of study. These results are consistent with those from Yassin et al. [1] and Heier et al. [9]. However, a trend was observed in which FWM decreases over the years, in contrast to what was observed in the Heier et al. [9] study. These authors did not expect this result because the two new breeds had been introduced in Norway two years before their study, and they hoped that the farmers were already familiar with the management of these genetically novel birds.

The trend of FWM annual decrease observed in the present study could be related to the implementation of more effective management and biosecurity measures in broiler farms. A major effort has been made in recent years to improve vaccination techniques in the hatcheries in order to improve and homogenize these practices and to achieve an adequate immunization of the chicks. As an example, field vaccination has been changed to hatchery vaccination. In addition, the equipment used for vaccination is becoming more precise and welfare conscious, decreasing the possible errors. Hatcheries have been improving the quality of facilities and equipment, as for example the expedition rooms with better air-conditioning systems or new equipment to supplement chicks in the hatcheries with probiotics and nutrients, which may be another reason why FWM decreased over the years. Therefore, it is expected that if both biosecurity and management improve, there would be a concomitant improvement in animal welfare reflected in performance indices such as lower mortality rates. However, the possible influence of genetics on these results was beyond the scope of this study and it could not be elucidated.

#### 4.2.5. Season

The highest FWM was detected in autumn and winter, whereas Yassin et al. [1] observed higher FWM rates during the spring and lower rates during autumn. These differences could be related to the geographical location. The study of Yassin et al. [1] was carried out in the Dutch poultry industry. The Dutch climate is temperate and very humid with narrower temperature ranges compared with the Spanish climate. The difference between summer and winter in Spain can be very extreme due to the influence of microenvironments.

The seasonality in mortality could be related to climate, since the lowest environmental temperatures are in winter. It should be considered that new-hatched chicks do not have a mature thermoregulation system, which do not become fully functional until day 7–10 of life [44]. Therefore, if chicks are exposed to low ambient temperatures due to improper handling, the chances of increasing FWM are up to three times greater [45]. Another reason for this seasonality is that it could be related to market fluctuation. When demand is high, the hatcheries buy a lot of external eggs. These eggs, however, are mostly of variable quality and can therefore result in lower viability of chicks.

Given the influence of seasonality on the FWM, it would be important to ensure that possible seasonal fluctuations in the environment are controlled and minimized as far as possible. Production companies should review the environmental and tightness conditions of all their hatchery rooms, broiler farms and transport with the aim of reducing the possible impact on FWM.

## 5. Conclusions

Results indicate that breeder age, chick gender, breed, type of broiler house, presence or absence of drip cup, egg storage, study year and season were factors that influenced first-week mortality, which confirms and builds on the findings from previous studies. Therefore, these factors should be key in the decision making of poultry breeding companies, in order to reduce possible welfare problems and increase productive performance.

Regarding the other factors detected, results suggest that they should also be considered. However, due to the variability, possibly caused by geographical location and human factors, more epidemiological studies are necessary.

## Figures and Tables

**Table 1 animals-10-00310-t001:** The list of independent variables classified as external (environmental or management) and internal (individual-specific) factors, and the total mean cumulative mortality during the first week of life (study of 2267 flocks in 104 farms and 253 broiler houses, Spain, 2015–2018).

Factor *	Category	N Farms	N Houses	N Flocks	Mean FWM ** (%)
Farm ID		104	253	2267	1.82
Province	Barcelona	3	7	60	1.76
	Huesca	39	95	880	1.87
	Lleida	34	99	847	1.69
	Tarragona	6	14	150	1.73
	Teruel	1	1	14	1.79
	Zaragoza	19	37	316	2.05
Age of the broiler house	<10 years	23	43	404	1.83
	10–20 years	27	56	555	1.70
	>20 years	67	152	1301	1.87
	Unknown	2	2	7	1.59
Heating system	Air generators	58	153	1419	1.79
	Heat screens	34	73	612	1.91
	Cannon	7	14	99	1.81
	Other types	7	10	117	1.69
	Unknown	3	3	20	1.57
Litter	Straw	8	24	193	1.71
	Shavings	18	40	341	1.89
	Rice hulls	70	173	1565	1.80
	Oat hulls	3	6	62	2.28
	Unknown	6	10	106	1.80
Farm entry density—chicks/m^2^	<17.5	26	65	611	1.95
	≤17.5–<30	23	54	442	1.74
	≤30–<36	32	69	668	1.71
	≥36	29	65	546	1.86
Study year	2015			591	1.93
	2016			612	1.75
	2017			610	1.80
	2018			454	1.79
Type of broiler farm	One floor	80	178	1951	1.83
	Multiple floors	30	73	309	1.75
	Unknown	2	2	7	1.82
Type of insulation	Bridge work	75	179	1544	1.81
	Sandwich panel	34	66	657	1.85
	Cosma	4	6	59	1.64
	Unknown	2	2	7	1.82
Feeder brand	Tigsa	15	28	257	1.74
	Chore-time	24	50	432	1.93
	Roxell	30	61	613	1.67
	Ska	24	47	429	1.89
	Other brands	22	47	393	1.93
	Unknown	10	20	143	1.73
Egg storage days	<7			688	1.94
	7–10			1368	1.76
	>10			209	1.76
	Unknown			2	1.70
Lighting	Light-emitting diode (LED)	14	26	218	2.10
	Non-adjustable fluorescent	56	135	1287	1.77
	Adjustable fluorescent	17	34	307	1.90
	Bulb	28	57	452	1.76
	Unknown	1	1	3	2.08
Season	Winter 22 December–20 March			523	1.93
	Spring 21 March–21 June			575	1.69
	Summer 22 June–22 September			600	1.81
	Autumn 23 September–21 December			569	1.86
Drinker brand	Lubing	17	28	229	1.74
	Plasson	18	42	414	1.70
	Corti zootecnici	63	143	1294	1.89
	Other brands	16	38	323	1.74
	Unknown	2	2	7	1.82
Drinker flow rate	High	55	112	1044	1.70
	Low	52	134	1161	1.92
	Unknown	5	7	62	1.88
Presence of drip cup	Yes	59	130	1152	1.66
	No	51	119	1083	1.98
	Unknown	59	130	32	1.92
Ventilation	Natural	6	18	116	1.47
	Tunnel	27	51	446	1.78
	Transverse	57	125	1044	1.83
	Transverse + Tunnel	24	55	630	1.92
	Natural + Tunnel	1	2	24	1.22
	Unknown	2	2	7	1.82
Fuel	Gas	74	163	1490	1.93
	Diesel	12	31	229	1.62
	Biomass	14	41	366	1.66
	Other types	7	16	175	1.47
	Unknown	2	2	7	1.82
Flock size	<6.700			1133	1.86
	≤6.700–<9.300			521	1.80
	≤9.300–<13.000			309	1.83
	≥13.000			304	1.78
Breeder age	25–33 weeks			541	1.79
	34–50 weeks			1292	1.76
	51–68 weeks			433	2.10
	Unknown			1	1.72
Chick gender	Mixed			149	1.91
	Male			962	1.89
	Female			1156	1.74
Breed	1			1717	1.85
	2			550	1.72

* External factors considered in this table were extracted from the following references: Yassin et al. [1], Vieira et al. [2] Peebles et al. [6], Heier et al. [9], Chou et al. [10], Thoghyani et al. [12] and Rozemboim et al. [13]. ** FWM – First week mortality.

**Table 2 animals-10-00310-t002:** The results from the generalized linear mixed model of the cumulative first-week mortality in the study of 2267 flocks in 104 farms with 253 houses (Spain, 2015–2018).

Type III Tests of Fixed Effects				
Factor	Num DF	Den DF	F Value	Pr > F
Type of broiler house	1	242.6	6.28	0.0129
Drip cup	1	230	23.67	<0.0001
Study year	3	6380	424.32	<0.0001
Season	3	6380	449.73	<0.0001
Egg storage days	2	6380	491.92	<0.0001
Breeder age (weeks)	2	6380	3210.88	<0.0001
Chick gender	2	6380	1762.58	<0.0001
Breed	1	6380	1350.97	<0.0001

**Table 3 animals-10-00310-t003:** The results of the cumulative first-week mortality in the study of 2267 flocks in 104 farms with 253 houses (Spain, 2015–2018) for the differences between each pair of categories of every significant variable (differences were measured in terms of OR).

Factor	Category a	Category b	*p*-Value	OR ^1^ (CI ^2^)
Type of broiler house	One floor	Multiple floor	0.0129	1.09 (1.02–1.15)
Drip cup	No	Yes	<0.0001	1.17 (1.10–1.25)
Season	Winter	Autumn	0.0027	1.01 (1.00–1.02)
	Winter	Spring	<0.0001	1.10 (1.09–1.10)
	Winter	Summer	<0.0001	1.05 (1.04–1.05)
	Autumn	Spring	<0.0001	1.08 (1.08–1.09)
	Autumn	Summer	<0.0001	1.03 (1.03–1.04)
	Summer	Spring	<0.0001	1.05 (1.04–1.05)
Study year	2015	2016	<0.0001	1.09 (1.08–1.09)
	2015	2017	<0.0001	1.06 (1.06–1.07)
	2015	2018	<0.0001	1.10 (1.09–1.10)
	2017	2016	<0.0001	1.02 (1.01–1.03)
	2016	2018	0.0107	1.01 (1.00–1.02)
	2017	2018	<0.0001	1.03 (1.02–1.04
Egg storage days	<7	7–10	<0.0001	1.07 (1.06–1.07)
	7–10	>10	<0.0001	1.03 (1.02–1.04)
	<7	>10	<0.0001	1.10 (1.09–1.11)
Breeder age (weeks)	34–50	25–33	<0.0001	1.04 (1.03–1.05)
	51–68	25–33	<0.0001	1.26 (1.25–1.26)
	51–68	34–50	<0.0001	1.22 (1.20–1.22)
Chick gender	Males	Females	<0.0001	1.11 (1.11–1.12)
	Mixed	Females	<0.0001	1.17 (1.17–1.18)
	Mixed	Males	<0.0001	1.05 (1.04–1.05)
Breed	Ross	Cobb	<0.0001	1.08 (1.07–1.08)

^1^ OR—Odds Ratio ^2^ CI—95% Confidence Intervals.

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
