# Peer review of "Factors Associated with Cumulative First-Week Mortality in Broiler Chicks"

_animals, 2020, doi:10.3390/ani10020310_

Round 1

Reviewer 1 Report

Please find my comments in the attached document. 

Author Response

Please find my comments in the attached document. 

Reviewer 2 Report

The study aims to identify internal and external factors that could influence the cumulative first-week mortality of broilers. The research objective is clearly presented. Researchers discussed the effect of external (management or environmental) and internal (individual-dependent) factors on mortality.

However, this meta-analysis study is based on mortality information provided by a hatchery company, the external and internal factors were not presented in the manuscript. The cobb chicken guide just provides a reference, the real environment of poultry production is varying depending on farm management. For improving the study, researchers need to verify your findings by using some farm data that are not used in your statistical analysis.

Other suggestions:

The writing needs significant improvements. Grammatical errors are observed in many places. Structure of writing needs considerable modification. For example, how can a section title cannot be called tables (see line 156: 3.1 tables)? section 2.2 needs further explanation by including more information such as different ex or inter factors (e.g., breed, age, gender, environmental factors, etc.) listed in guideline or other publications. References was added for information in Table 1. Where did you get the information? Structure of table needs to be reconstructed.

Author Response

(The authors gave the same response as above.)

Round 2

Reviewer 2 Report

The writing was improved significantly as compare to the last version. There are still a number of major questions in the current version:

The data requested from farm or cited from other references such as flock, gender and broiler house (or half broiler house. The study year is reported with significant effect on mortality, this need to be discussed further? Authors reported significant factors only, what about other factors such as feeding, bedding, and air quality? Any factors were found with insignificant factors? How accurate your estimates are? Verification is needed.

Round 3

Reviewer 2 Report

This version was improved as compared to the last one, but authors addressed some primary suggestions in a very brief way such as air quality and ventilation.

In broiler production system, the brooding environmental control (e.g., litter moisture, air temperature and relative humidity, air quality) is critical for mortality reduction. Authors should include those sides into the analysis.

Besides, authors should discuss more on measures that may reduce mortality for producers' consideration. 
